# Systemic Agent-Based Modeling and Analysis of Passenger Discretionary Activities in Airport Terminals

**Adin Mekić, Seyed Sahand Mohammadi Ziabari * and Alexei Sharpanskykh**

Air Transport and Operations, Faculty of Aerospace Engineering, Delft University of Technology, Kluyverweg 1, 2629 HS Delft, The Netherlands; A.Mekic@tudelft.nl (A.M.); O.A.Sharpanskykh@tudelft.nl (A.S.)

* Correspondence: s.s.mohammadiziabari@tudelft.nl

**Abstract:** Discretionary activities such as retail, food, and beverages generate a significant amount of non-aeronautical revenue within the aviation industry. However, they are rarely taken into account in computational airport terminal models. Since discretionary activities affect passenger flow and global airport terminal performance, discretionary activities need to be studied in detail. Additionally, discretionary activities are influenced by other airport terminal processes, such as check-in and security. Thus, discretionary activities need to be studied in relation to other airport terminal processes. The aim of this study is to analyze discretionary activities in a systemic way, taking into account interdependencies with other airport terminal processes and operational strategies used to manage these processes. An agent-based simulation model for airport terminal operations was developed, which covers the main handling processes and passenger decision-making with discretionary activities. The obtained simulation results show that operational strategies that reduce passenger queue time or increase passenger free time can significantly improve global airport terminal performance through efficiency, revenue, and cost.

**Keywords:** discretionary activities; airport terminal operations; agent-based modeling; operational strategies

## 1. Introduction

The airport terminal is a complex sociotechnical system that includes multiple processes and activities in which passengers engage with operators. These activities are divided into mandatory activities such as check-in and security, and discretionary activities such as retail. Generally, mandatory activities are studied by formal and computational modeling [1], whereas discretionary activities are rarely taken into account. However, discretionary activities play an important role in airport terminals, as they affect passenger flows and are a major component of revenue generation. For example, retail, food, and beverages constitute of approximately 35% of non-aeronautical revenue [2], an important growing source of income for the global aviation industry. Since discretionary activities have a large influence on airport terminal performance they need to be studied in detail. This paper focuses on formal modeling and analysis of such activities. Furthermore, discretionary activities are largely influenced by and related to mandatory airport terminal activities, such as security check. These mandatory processes may impact dwelling time and free time, which passengers can use for discretionary activities. Moreover, there are many other relations between the terminal processes. In particular, the check-in process influences passenger arrival at security checkpoints, and check-in and security processes influence passenger dwelling time, which impacts commercial activities and passenger gate arrival. Therefore, discretionary activities need to be studied in relation to all these processes. In this paper, we take a systemic approach to airport terminal modeling.

Discretionary activities are pursued based on passengers' freedom of choice, and decision-making. Decision-making is a complex cognitive process, and is influenced by passenger traits, time availability, affective states, perception, and anticipation. Existing

research related to modeling of discretionary activities at airport terminals does not attempt to formalize a cognitive framework for passenger decision-making [3–5]. Instead, probabilistic models are used to directly specify passenger decision-making on discretionary activities. An advantage of such a high-level modeling framework is the simplicity of calibration of the formal model. However, such a framework does not represent realistic human reasoning and decision-making. Therefore, detailed cognitive modeling of passengers is required to represent discretionary activities.

Discretionary activities are directly influenced by multiple processes, where each process can be managed using diverse operational strategies. Check-in counter allocation [6,7], security checkpoint lane allocation [8,9], virtual queuing [10], and call-to-gate are examples of airport terminal operational decision-making problems in which multiple strategies can be considered. For example, security checkpoint lane allocation specifies the number of open lanes per time interval. Call-to-gate strategies delay presenting passengers the gate allocation to increase passenger engagement with retail, food, and beverages [11]. Due to process dependencies, different strategies for a specific process influence global airport terminal performance.

To analyze and model the airport terminal processes, agent-based modeling and simulation was used. It allows for detailed modeling of large-scale complex adaptive sociotechnical systems with many heterogeneous components such as humans with their cognitive processes. Moreover, it enables analyzing relations between local processes, and analyzing emergent properties of the global system. Agents can be used to accurately represent passenger interaction with the airport terminal environment. Thus, agent-based modeling is a suitable paradigm to model all airport terminal processes, including passenger interaction with discretionary activities.

The goal of this research is to analyze how different call-to-gate and security lane allocation strategies impact passenger engagement with discretionary activities and airport terminal performance. In this study, airport terminal performance is measured by queue time, expenditure generation, missed flights, and operational cost. To achieve this, an existing agent-based model of airport terminal operations was expanded to include passenger decision-making about discretionary activities. Another important contribution of the paper is a cognitive model which handles passenger decision-making concerning discretionary activities. In this model, passenger anticipate on future events and perform planning of their activities in an airport terminal. Furthermore, passenger characteristics are taken into account, as they influence the decision-making and planning processes of agents. The developed cognitive model is based on affective decision-making theories from behavioral sciences.

The paper is organized as follows. Related work is presented in Section 2. Then, the methodology is introduced in Section 3. The agent-based model is described in Section 4. Section 5 presents the experiments and results. Section 6 discusses the broader implications of the study. Section 7 is the conclusion.

## 2. Related Work

This section presents a literature review which describes several topics that are relevant for passenger decision-making about discretionary activities. Section 2.1 reviews other research which considered shopping as a part of their airport terminal models. Section 2.2 describes how passenger characteristics influence passenger behavior at the airport terminal. Section 2.3 describes theories that are relevant for modeling passenger decision-making. Finally, Section 2.4 discusses operational strategies for airport terminal processes.

### 2.1. Airport Terminal Shopping Modeling

This section presents studies which have implemented shopping behavior in their airport terminal models. We specifically elaborate on passenger decision-making in these models.

Chen et al. use an agent-based model with shops and restaurants to investigate the impact of terminal design on retail performance [4]. Passenger characteristics determine

the decision-making. Although a decision-making logic is presented, this work does not provide a cognitive framework for decision-making. Kleinschmidt et al. use an agent-based model to include duty-free shopping at the airport terminal [5]. The model does not include complex decision-making of agents, but rather a fixed probability is used to determine what passengers do at decision points. The use of decision-points limits the ability for agents to make decisions at any time point. Since the airport terminal environment is dynamic, passengers should be able to make decisions under changing conditions. Ma et al. use an agent-based model where Bayesian networks define the conditional probabilities for passengers to use airport terminal facilities based on their personal traits [3,12]. Many different discretionary activities are implemented such as shopping, purchasing food, asking for assistance, use of kiosks, money withdrawal, resting, and more. The environment in the model is another component that impacts passenger decision-making. Similarly, in this study decision-points are also used. Agents at decision-points decide stochastically about target and route choice, and determine choices based on the utility maximization principle. Furthermore, Bayesian networks are extended with influence diagrams to model agent beliefs and available actions. Available time and walking distance influences if a passenger decides to pursue an activity. For example, if time until departure is below a certain threshold, passengers will not pursue activities. A limitation in this model is the lack of a planning framework for agents. Agents decide to pursue the activity that provides most utility one at a time, thus there is no evaluation of a sequence of activities. Thus, passengers are not able to anticipate future actions.

The described studies all use agent-based modeling and simulation. Because they lack a cognitive architecture, they do not incorporate a method for sequencing actions, and ignore that agents are able to anticipate the consequences of their actions. Thus, there is a lack of human realism in the decision-making.

### 2.2. Airport Terminal Passenger Characteristics

Passenger characteristics influence how passengers interact with discretionary activities. Chen provides an extensive literature review on factors that influence retail revenue [13]. Age, gender, and travel purpose are commonly used to segment passengers in relation to discretionary activities. Older passengers generally shop less than the average passenger, but spend more per purchase as they have higher purchasing power than younger passengers [14,15]. Females spend more time shopping, browse more, purchase more, and have more planned purchases than males [16–18]. Compared to leisure passengers, business passengers interact less with retail because they spend less time at the airport, and therefore spend less money. Furthermore, traveling companions influence purchase behavior. For example, passengers traveling in groups spend more money [13,18], and wavers and non-ticket holding companions impact passenger purchasing behavior in the pre-security area [19].

Alternative methods exist to segment passengers. Chung et al. specify four different types of shoppers: apathetic shoppers who do not interact with shops, traditional shoppers who pre-plan their purchases, mood shoppers who interact with shops depending on the atmosphere, and shopping lovers who highly interact with shops [20]. Harrison et al. suggest that basic criteria, such as travel purpose and travel frequency, do not provide enough insight into the passenger experience [21]. They segment passengers based on degree of engagement and sensitivity of time. Based on this, three passengers groups can be distinguished: airport enthusiasts who arrive early at airports and actively engage in discretionary activities, time fillers who arrive early at airports but do not engage with activities, and efficiency lovers who arrive later at airports and do not engage with activities. Combined with basic criteria such as age, gender, and travel purpose, this classification can be used for the detailed modeling of passenger behavior related to interaction with discretionary activities. For example, Ma provides conditional probability tables that specify probabilities for goal generation based on passenger characteristics [12].

Note however that all airports behave differently, and similar studies at different airports can result in very different outcomes. For example, the average expenditure in a medium-sized Spanish airport was found to be EUR 4.25 [22], whereas in a large-sized Taiwanese airport average expenditure was found to be TWD 88.5 [23]. This can partially be attributed to passenger characteristics, as the airports process different types of passengers.

*2.3. Decision-Making Theories for Passenger Modeling*

Understanding decision-making of passengers is essential for the modeling of airport terminal operations, and in particular for passenger engagement with the discretionary activities. Multiple factors impact passenger decision-making, such as available time, emotion, perception, anticipation, and passenger characteristics. This section elaborates on these factors, and presents theories that could be implemented to model complex passenger decision-making at the airport.

Decision-making generally involves generating and evaluating alternatives that represent an action, or sequence of actions. Traditionally, evaluating alternatives was performed based on utility theory. However, the classical utility theories are not able to represent realistic human reasoning as they do not take into account biases and affective states, which influence human decision-making [24–26]. Including affective states in models of humans in the context of airport terminals could be essential, since passengers experience feelings such as anxiety, stress, and excitement [27]. The OCC (Ortony, Clore, and Collins) framework has been frequently used to model emotions in human agents [28,29]. It describes a hierarchy of emotions and mechanisms of emotion generation.

Action alternatives are being evaluated by the human brain by forward simulating the consequences of the actions [30]. Hesslow formulated this in the simulation theory of cognitive function [30], and provides evidence that simulated behavior is very similar to actual behavior. The following assumptions are made in the simulation hypothesis [30]:

1. *"Simulation of actions: we can activate motor structures of the brain in a way that resembles activity during a normal action but does not cause any overt movement."*
2. *"Simulation of perception: imagining perceiving something is essentially the same as actually perceiving it, only the perceptual activity is generated by the brain itself rather than by external stimuli."*
3. *"Anticipation: there exist associative mechanisms that enable both behavioral and perceptual activity to elicit other perceptual activity in the sensory areas of the brain. Most importantly, a simulated action can elicit perceptual activity that resembles the activity that would have occurred if the action had actually been performed."*

External stimuli elicit perceptual activity in the brain, which leads to response preparation before an action. The mechanism of anticipation allows for response preparation states to elicit new perceptual activity, which leads to new response preparations. Thus, internally generated perceptual activity forms perception chains of behavior, which represent a simulated sequence of actions and their consequences. This concept can be used in computational models in which human agents evaluate different sequences of action to create a planning. Sharpanskykh and Treur applied the simulation theory to an evacuation scenario, where passengers made choices between paths [31]. Since emotion is an important component in human decision-making under time pressure and uncertainty, this work also provides a framework that describes how emotion elicitation could be included as a part of a cognitive decision-making model.

Airport terminal passenger decision-making requires the modeling of the passenger's perception of time, since time highly influences passenger actions at the airport. Passengers have incomplete information about duration of future events, which requires them to make prospective time estimations about future events to anticipate their time availability. If passengers perceive themselves to have available time, they may pursue goals related to discretionary activities.

Passenger characteristics also influence passenger emotions and decision-making processes. The five-factor model is often used to specify the personality of agents, and

consists of openness, conscientiousness, extraversion, agreeableness, and neuroticism [32]. Considering airport terminal processes, passengers scoring high on neuroticism can be more prone to impulsive buying [33], and are more prone to trait anxiety. Considering that prospective time estimates play an important role in decision-making at the airport terminal, prospective time estimates are overestimated for people scoring higher on anxiety [34], or scoring higher on extraversion [35]. The overestimation can be attributed to the higher arousal, which increases the pace of the internal clock mechanism [36]. It has also been argued that an overestimation is not an incorrect functioning of the internal clock, but rather an adaptation from the internal clock towards the environment [37].

*2.4. Airport Terminal Operational Strategies*

This section presents strategies that impact passenger flow and essential performance indicators of the airport terminal. We discuss how these strategies impact the performance of an airport terminal.

The check-in counter allocation strategies aim to balance operative costs and passenger waiting times at the airport terminal check-in. Delays at check-in can result in delays in other systems such as security checkpoint or flight departure. Various modeling studies examined how cost and efficiency performance can be optimized for check-in counter allocation [6,7]. A similar resource allocation problem, to optimize operational costs and waiting times, exists for security checkpoint lanes. Again, existing studies suggested different strategies on how this problem can be optimized [8,9]. These strategies may maximize for efficiency, minimize for cost, or optimize for a combination of the two. Since airport terminal processes depend on each other, existing studies have also addressed some of these dependencies. For example, Adacher showed how resources could optimally be allocated for both check-in and security checkpoint [38].

Virtual queuing is a method to provide time slots to passengers to conduct an activity. De Lange showed that virtual queuing at airport security lanes can reduce operating costs and passenger queue time [10]. However, its success depends on the reliability of the forecast model and passenger arrival pattern. A disadvantage is that passengers have to wait in the pre-security area for long periods of time. Although this could increase average dwelling time, it could reduce retail revenue as passengers would spend less time in the central shopping area, which in most airport terminal layouts is after the security. More studies need to be conducted on the opportunities of improving airport performance due to virtual queuing. For example, the changes of performance due to virtual queuing at check-in have not yet been analyzed.

Call-to-gate strategies were also investigated, but to a limited extent. Moulds et al. mentioned that call-to-gate strategies can relax passengers because they are informed directly through their mobile phone, and increase retail revenue as passengers are more at ease and have more dwelling time in the shopping areas [11].

## 3. Methodology

Figure 1 shows the steps that were taken to fulfil the research objective. The existing Agent-based Airport Terminal Operation Model (AATOM) [39,40], that includes the main passenger handling processes, was extended to include complex passenger decision-making about discretionary activities. AATOM contains passenger agents, operator agents, and orchestration agents [40].

The first step is to define the mandatory airport terminal activities of the airport terminal layout. This includes mandatory airport terminal activities such as check-in, security checkpoint, passport control, and gate procedure. These activities are already components of AATOM and can be used and altered to define the specific airport terminal activities.

The second step is the definition of the discretionary activities. Discretionary activities are the optional activities that an agent can pursue, specifically, activities related to restaurants and shops need to be defined. Agent goals related to discretionary activities can be generated either when the agent is spawned or after that.

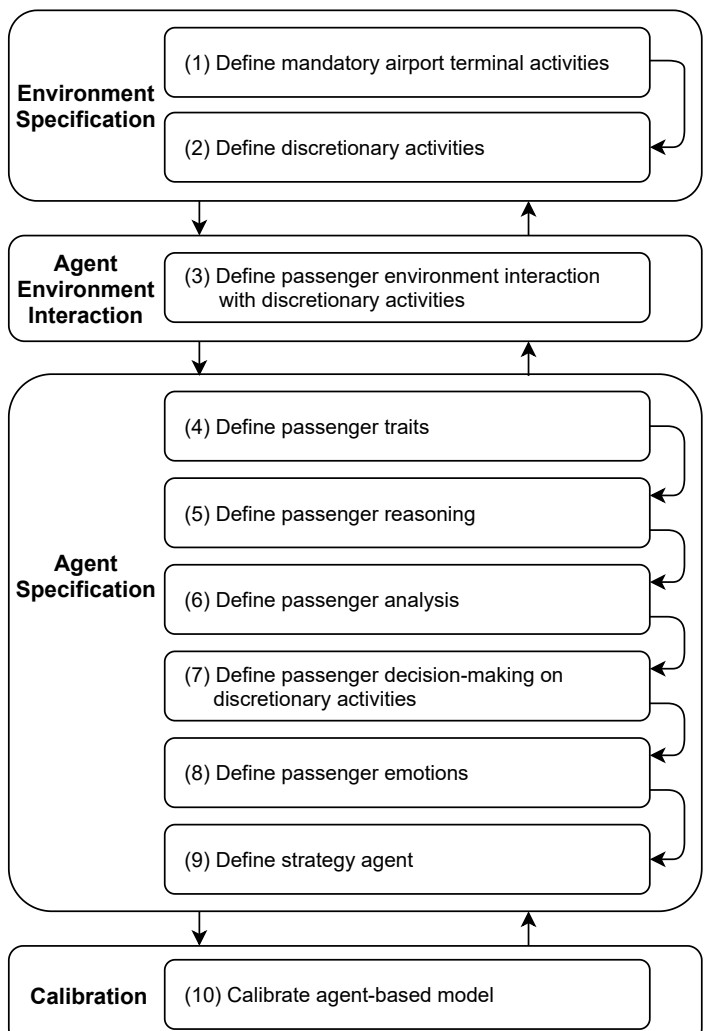

**Figure 1.** Methodology overview.

The third step is to define the interaction of passengers with the environment. This defines the actions of passenger agents related to the discretionary activities. AATOM interaction of passenger agents with check-in, security checkpoint, and gate processes is described by Janssen [39,40].

The fourth step is to define the passenger traits. Depending on their traits, passengers interact differently with discretionary activities. Basic traits can be age, gender, and flight purpose. Personality traits such as extraversion and neuroticism could also be defined. Traits can influence internal states, for example, how passengers perceive their environment and make decisions.

The fifth step is to define the reasoning architecture of the passenger agent. Analysis, decision-making, emotions, planning, and their relations form the reasoning process of the passenger agent. The reasoning architecture provides also the ability for passengers to evaluate possible future states when making decisions.

The sixth step is to define the analysis module of the passenger agent. By means of analysis, passenger make prospective time estimations. Since involvement of passengers in discretionary activities largely depends on their dwelling time, passengers have to determine how much free time they have. This perception can be influenced by their traits. When passengers arrive at the airport, uncertainty of their prospective time estimation of future events will be high. Moving through the airport, uncertainty decreases since the time estimates are updated based on the passenger observations. The sum of all prospective time estimates can be used as an indication of how much free time a passenger has.

The seventh step is to define passenger decision-making on discretionary activities. The decision-making depends on the passenger goals and the contextual and internal conditions, such as available time, location, and emotion. Passenger traits can also influence how decisions are made. Decisions made are provided as inputs to the planning module of the passenger agent.

The eighth step is to define how passenger agent's affective states, such as anxiety and excitement, influence other modules. Affective states depend on internal states and traits of passengers. For example, excitement is an affective state that can exist when discretionary activities are performed, or anxiety can exist if completion of important goals is at risk.

The ninth step is to define a special agent type, called the strategy agent. The strategy agent controls all the possible strategies related to the airport terminal processes.

The final step is to calibrate the agent-based model so that the parameters are configured to represent reality as accurately as possible.

## 4. Agent-Based Model

The proposed agent-based model is an extension of the AATOM baseline model with representations of discretionary activities and passenger decision-making about them. The cognitive architecture of the AATOM model is seen in Figure 2. To represent decision-making about discretionary activities, reasoning and activity modules were extended and agent traits were introduced. The extended and improved blocks are shaded in Figure 2.

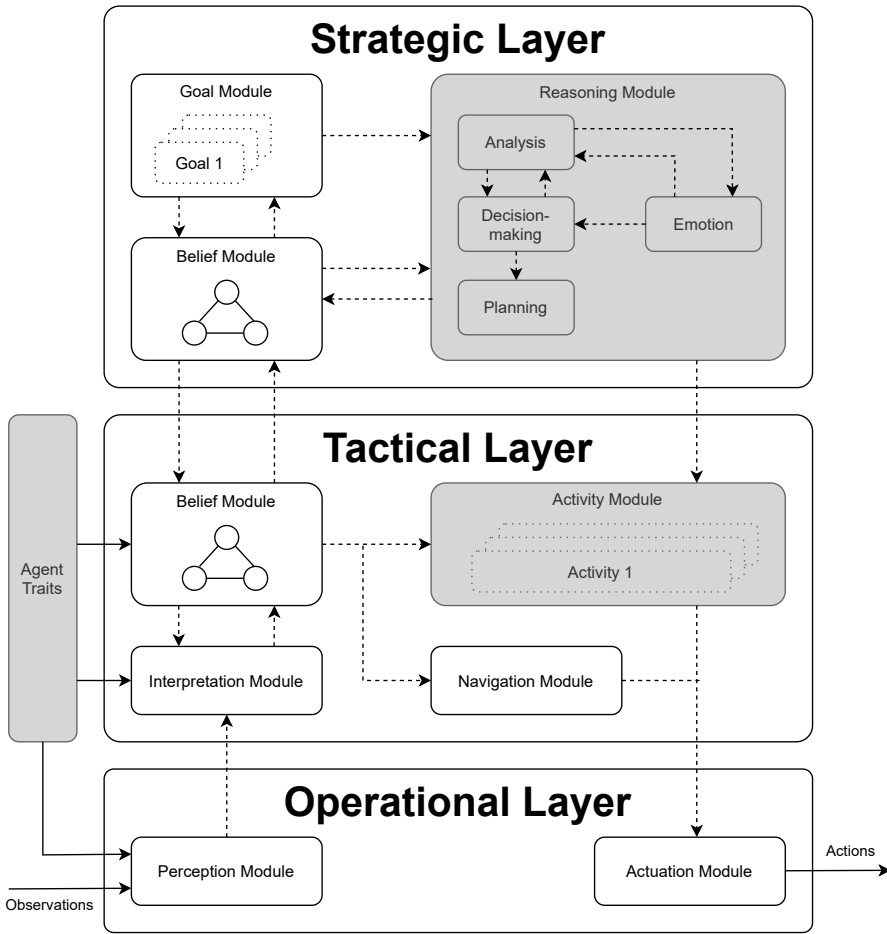

**Figure 2.** AATOM architecture adapted and improved from [39,40].

A passenger agent has a set of goals that drive its behavior. The passenger agent makes observations about the environment, which are processed in the interpretation module. The passenger agent then performs a reasoning process, which consists of analysis, decision-making, and planning. Analysis consists of analyzing the environment for condi-

tions required for decision-making with respect to discretionary activities, decision-making consists of making decisions about discretionary activities when their conditions are met, and planning generates a plan that satisfies the passenger agent's goals. The reasoning process is influenced by the passenger agent's emotions and beliefs, which may be influenced by the passenger agent's traits. The subsequent steps carry out activity control, actuation, and performing the actions to complete the activities. In the following subsections the steps of the methodology are elaborated in more detail.

### 4.1. Mandatory Airport Terminal Activities

Check-in, security checkpoint, and gates, are modeled as areas related to mandatory airport terminal processes. In AATOM, areas are objects, and are shapes accessible to a all human agents [39,40]. Check-in areas contain queues, desks, and check-in operators which check passengers in. Security checkpoint areas consist of queues and checkpoint lanes. Checkpoint lanes have belts, a walk-through metal detector, and an X-ray sensor. Security checkpoint areas contain X-ray operator, physical check operator, and luggage check operator agents. X-ray operators operate based on the X-ray sensor output. Physical check operators perform physical checks if the walk-through metal detector alarms. Luggage check operators perform luggage checks on luggage that is detected by the X-ray operator. The specification of the operator agents is provided in [40]. Gate areas may contain seats which passengers can use waiting before flight departure.

### 4.2. Discretionary Activities

Shops, restaurants, and restrooms are facilities modeled as areas, and are part of performing discretionary activities. Shops include operator agents at which passenger agents can make purchases, and may include walls to incorporate passenger queuing before a purchase activity. Restaurants are areas that include a number of seats, and an operator agent at which passenger can make purchases. The number of seats defines the capacity of the restaurant. Walls may be included to incorporate passenger queuing before a purchase activity. Restrooms do not contain other non-agent objects.

### 4.3. Passenger Agent Interaction with Environment

A total of five different discretionary activities were modeled: planned shop activity, shop browse activity, general shop activity, restaurant activity, and restroom activity. Multiple shop activities were modeled since passengers interact with shops in different ways. Activity goal generation and interaction with facilities is probabilistic, and may be defined by conditional probability tables related to passenger traits, described in Section 4.4.

Planned shop activity is a shopping activity in which the passenger agent moves to a single shop to make a purchase. The goal for this activity can be generated when the agent is spawned. Passengers who know beforehand that they want to buy a souvenir at the airport terminal are examples of planned shop activities. Planned shop activities are prioritized over other shopping goals in the planning module.

Shop browse activity is a shopping activity in which a passenger agent moves through the central shopping area that contains multiple shops. Passenger agents decide for each shop whether or not they want to enter. If a passenger agent enters a shop and spends some time there, a probability exists that the passenger makes a purchase. The shop browse goal can be generated when the passenger agent is spawned. This means the passenger agents know that they wants to browse the shops prior to airport arrival. A second decision-point is modeled after the security checkpoint to allow passengers to generate this goal if they want to spend their free time shopping instead of waiting. Shop browsing has lowest priority compared to other discretionary activities in the planning module.

General shop activity is an impulsive unplanned shopping activity in which a passenger interacts with the shop that is observed. Each time a passenger with dwelling time moves in close proximity of a shop, a probability exists for a passenger to enter the shop. The difference with other shopping activities is that this activity is not planned in advance,

thus the goal is generated when the passenger agent decides whether or not it wants to visit the shop. If the goal is generated, the activity is set as the first activity in the sequence of activities in the passenger planning.

Restaurant activity is an activity in which the passenger agent moves to a restaurant, makes a purchase, and sits on a chair for some time. Generation of the restaurant goal is probabilistic and generated when the agent is spawned. A distinction is made between pre-security and post-security restaurant goals and activities. Pre-security restaurant goals may be generated if a passenger has a waver or a non-ticket holding companion [19]. Restaurant goals are prioritized over shopping goals in the planning module.

Restroom activity is an activity in which the passenger agent moves to a restroom and spends some time. A goal for this activity can impulsively be generated each timestep in the goal module. For each passenger agent, the probability of a restroom visit is $\frac{1}{180}$ per minute with the assumption that humans visit a restroom once every three hours on average. If the goal is generated, the activity is planned after the current activity is completed. If the passenger is waiting at a gate, the gate activity is paused to activate the restroom activity.

### 4.4. Passenger Agent Traits

Passenger agents have five traits: passenger type, age, gender, class, and waver. Age, gender, and class are the typical basic traits that influence how passengers interact with facilities.

The first trait is the passenger type. Types define the level of engagement with discretionary activities, and the time sensitivity to the airport [21]. Time sensitive passengers are bothered by queuing and waiting [21]. Three passenger types are specified in the model: *airport enthusiasts* (AE) have a high level of engagement with low time sensitivity, *time fillers* (TF) have a low level of engagement with low time sensitivity, and *efficiency lovers* (EL) have a low level of engagement with high time sensitivity. Thus, efficiency lovers arrive later at the airport compared to time fillers and airport enthusiasts. Airport enthusiasts have high probabilities of interacting with shops and restaurants, whereas time fillers and efficiency lovers rarely interact with discretionary activities.

The four other traits are known as basic traits. The basic traits specify the age, gender, class, and waver. These traits are selected because they influence interaction with facilities. Class indicates leisure or business passenger, and waver indicates whether the passenger has non-traveling company. Age, gender, and class determine the movement speed of the passengers [41]. A waver increases probability for pre-security restaurant goal generation [19]. Based on the identified traits, conditional probability tables can be defined that specify, goal generation probability, enter probability, purchase probability, and expenditure distribution, for each specific facility based on traits that are defined above. In this case study, the generation of discretionary goals was only conditioned on passenger type, defined in Table A1. Shop browse enter probability, shop purchase probability, and shop purchase expense were assumed to be constant for all passengers and facilities, provided in Table A1.

### 4.5. Passenger Agent Reasoning

The passenger reasoning process consists of analysis, decision-making, emotion, and planning, as seen in Figure 2. Reasoning starts with analysis about the environment, which includes prospective time estimations of future actions, described in Section 4.6. Then, the decision-making module uses the analysis output to make decisions on future actions. These decisions are then incorporated into the planning. Emotions are an input to both analysis and decision-making and can influence prospective time estimations and decisions.

The architecture of reasoning is based on behavior and perception chains [30,31], described in Section 2.3, which allows for realistic modeling of human decision-making. The ability of agents to simulate future behavior is incorporated in the reasoning module.

Simulation of behavior and perception chains, seen in Figure 3, drives decision-making and therefore planning. Here, $s_n$ is the sensory state of action $n$, $r_n$ the response preparation state of action $n$, and $t_n$ is the time duration from $r_n$ to $s_{n+1}$. Prospective time estimations are used as links between perceptual activity and response preparation states. Thus, $t_n$ is a prospective time estimation an agent makes of action $n$. Simulated sensory states can elicit emotions which influence decision-making behavior [30]. This is addressed in Section 4.8.

In application to passengers in the airport terminal domain, the behavior and perception chains represent an evaluation of passenger's planning. The main passenger's goal is to catch their flight. Thus, passenger agents evaluate whether optional activities could fit in their planning without increasing risk of missing their flight. If a passenger perceives that an optional activity does increase this risk, the simulated activity will induce state anxiety in the chains, which drives the decision not to pursue this activity.

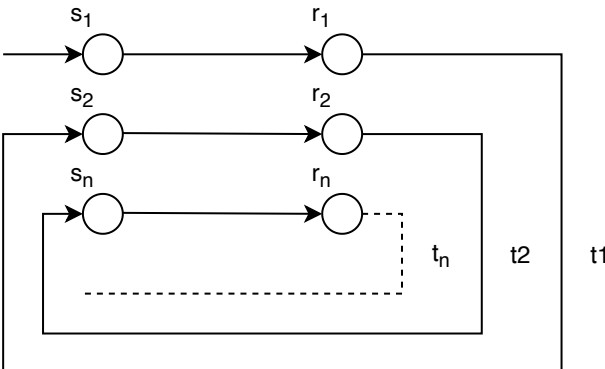

**Figure 3.** Behavior and perception chains.

*4.6. Passenger Agent Analysis*

The passenger analysis module consists of prospective time estimates for all activities and processes. At arrival, passengers make an initial prospective time estimation of the check-in, security, and time to walk to the gate. Initial estimates for these processes are assumed to be 10 min, 20 min, and 5 min, respectively. Prospective time estimates of check-in and security are updated when passengers are in close proximity of the queues. Observing the number of passengers in the queue allows for a more accurate prospective time estimate. The initial estimate is then updated by a more accurate estimate specified by $\frac{Q}{T*N} + C$, where $Q$ is number of passengers in the queue, $T$ is the passenger throughput per minute for a single desk or lane, $N$ is the number of open desks or lanes, and $C$ is a contingency factor in minutes. $T_{ci} = 1$ and $T_{sc} = 2$ are set from airport data [42,43]. $C_{ci} = 3$ and $C_{sc} = 5$ are assumed for all passengers, which requires the activity to be in the passenger's planning. Otherwise, $C_{ci} = 0$ and $C_{sc} = 0$.

Prospective time estimates on discretionary activities are decisions determined by realizations of random variables, defined for each discretionary activity in Table A1. This determines how much time a passenger spends during a discretionary activity. If that realization exceeds the passenger's available time, it is set to available time.

Total time, $T_t$, is the sum of all prospective time estimates of a passenger agent. It represents the passenger's time expectation to complete all existing goals. $T_t$ is impacted by affective states, described in Section 4.8. Total planned time, $T_p = T_t + \epsilon$. Here $\epsilon$ is a contingency factor accounting for uncertainty in the total time estimation. A passenger at the airport terminal entrance has high time estimation uncertainty, whereas at the gate uncertainty is low.

Free time, $T_f$, is the indication of how much free time a passenger has. $T_f$ is the main parameter passengers use to make decisions about their discretionary activities. $T_f = t_g - T_p$, where $t_g$ is the time left until the gate opens. It is assumed that passengers know when the gate opens, that passengers want to move to the gate when the gate opens, and that $T_f = 0$ once passengers arrive at the gate.

When passenger agents perform activities, the time spent at an activity is subtracted from its prospective time estimate. If a passenger agent completes an activity, and the time spent was less than planned for, the passenger agent's $T_f$ increases by the difference between spent time and planned time. When the passenger agent exceeds the planned time for an activity, $T_f$ starts to decrease.

### 4.7. Passenger Agent Decision-Making on Discretionary Activities

To make the decisions to pursue a discretionary activity, a goal for that activity is required to be activated, and for $T_f$ to be above a positive defined threshold. If these conditions are met, the decision is made to pursue that activity, and a prospective time estimate is drawn from a duration distributions, specified for each discretionary activity in Table A1. The free time estimate is updated by subtracting the time of the new activity, and the activity is added in the passenger's planning. If multiple discretionary goals exist, decisions are made sequentially in the order of importance. The activities are ordered according to the level of importance as follows: planned shop activity is the most important, then restaurant activities, next shop browse activity, and general shop activity is the least important. Restroom activities are directly included in the planning if the goal exists and the boarding process is not initiated, and therefore bypasses the decision-making module. Decisions can be made to cancel a discretionary activity depending on emotional states of the passenger agent.

### 4.8. Passenger Agent Emotions

Emotions influence passenger analysis and decision-making. The intensity of the affective states determines to what extent the analysis and decision-making are impacted. According to the OCC framework, intensity of prospect-based emotions such as fear and hope is based on: (1) the degree to which an event is desirable/undesirable, and (2) the likelihood of an event [28].

Anxiety is the only passenger agent emotion yet modeled in the cognitive architecture, but the architecture allows modular expansion of other emotions passengers experience at the airport terminal, such as excitement. Anxiety states are generated when a passenger perceives that it can miss a flight. For state anxiety, intensities are defined as low, medium, or high, which depend on the magnitude of $T_f$. Low intensity exists if $T_f > -5$, medium intensity if $T_f <= -5$, and high intensity if $T_f <= -20$. A low intensity does not influence other modules. A medium intensity increases prospective time in the analysis module by 10%, and cancels low importance discretionary activities such as the shop browsing and general shop activity. A high intensity increases prospective time estimates in the analysis module by 20%, and cancels all discretionary activities. Thus, affective states may remove discretionary activities that exist in the behavior and perception chains to move to a better emotional state.

### 4.9. Strategy Agent

The strategy agent controls time-based events that define simulation scenarios. The actions performed at a set of time points are considered as the strategy for a specific airport terminal processes. Therefore, strategies impact passenger flow, and influence $T_f$.

A total of five strategies are defined:

1. Check-in desk allocation specifies the number of open desks per time interval, for each check-in area.
2. Security checkpoint lane allocation specifies the number of open lanes per time interval.
3. Check-in virtual queuing specifies a strategy that defines time windows in which passengers should check-in.
4. Security checkpoint virtual queuing specifies a strategy that defines time windows in which passengers should move through the security checkpoint.
5. Call-to-gate specifies the time before flight departure at which passengers are called to the gate.

Check-in desk allocation, security checkpoint lane allocation, and call-to-gate strategies, are used in the case study.

*4.10. Calibration of the Agent-Based Model*

The model was calibrated using airport data from multiple sources, provided in Table A1 of Appendix A. *Simulation parameters* part contains parameters related to the simulation setup, including flight related parameters.

*General model parameters* part mainly contains probabilities and duration distributions. The security and check-in processes were calibrated to represent reality at a regional airport. Parameters related to the security process were calibrated using a passenger dataset [43]. The check-in parameters were calibrated using airport data [42]. Additionally, AATOM features such as check-in and security processes have previously been validated in AATOM case studies conducted by previous researchers [39]. In Table A1, $N(\mu, \sigma)$ represents a normal distribution with mean $\mu$ and standard deviation $\sigma$. $LogN(\mu, \sigma)$ represents a lognormal distribution with mean $\mu$ and standard deviation $\sigma$. $\Gamma(\alpha, \beta)$ represents a gamma distribution with shape $\alpha$ and scale $\beta$. Furthermore, arrival interval parameters specify the interval in which passengers arrive before flight departure. An arrival interval consists of three equally divided sub-intervals for which arrival distribution indicates the probability for a passenger to arrive in a specific sub-interval. *Trait model parameters* part contains probabilities that specify how passengers are generated. *Discretionary goal generation parameters* part contains probabilities for a passenger to create a goal. These parameters were defined under the assumption that airport enthusiasts perform more discretionary activities. *Decision-making model* parameters contain thresholds that specify the required $T_f$ to pursue a discretionary goal. Furthermore, the activity duration distributions, and probabilities for shop entering and purchasing are provided. The assumptions for the restaurant and shopping durations were based on data that indicates total shopping and dining time of passengers [18]. The restroom durations were based on data that indicate observed restroom dwell times of passengers [44]. All parameter assumptions related to restaurant and shop discretionary activities were based on expert knowledge obtained through interviews from commercial managers of a regional airport.

## 5. Experiments and Results

This section shows how the developed model was used for analysis and operational decision-making. The scenarios are described in Section 5.1. The experimental setup is presented in Section 5.2. The results of the call-to-gate and security lane allocation experiments are discussed in Sections 5.3 and 5.4, respectively. Section 5.5 shows how call-to-gate strategies are influenced by security lane allocation strategies.

*5.1. Scenarios*

A call-to-gate strategy determines the time at which passengers are called to their gate before flight departure time. Call-to-gate strategies impact passenger dwelling time, which influences passenger engagement with discretionary activities. Call-to-gate strategies in the developed model provide guidance when passengers should come to the gate, but does not withhold passengers from moving to the gate, since the gate number is known in advance. The goal of experiment 1 was to explore how an airport's call-to-gate strategy impacts passengers' expenditure and missed flights. For this experiment, different call-to-gate strategies were used, while strategies for other processes, as defined in Section 4.9, were fixed.

In the second experiment, a security lane allocation strategy was considered, which also influences passenger engagement with discretionary activities. The goal of experiment 2 was to explore how security lane allocation impacts security checkpoint average queue time, expenditure, and security checkpoint costs. Security operator cost was used to calculate security checkpoint cost. For this experiment different security lane allocation strategies were used, while strategies for other processes were fixed.

The third experiment is defined to explore how call-to-gate strategies are influenced by security lane allocation strategies. Depending on the call-to-gate strategy, importance of passenger throughput at the security checkpoint for time intervals may change, which can impact expenditure performance. For this experiment, the expenditure impact was analyzed with respect to call-to-gate strategies for the selected security lane allocation strategies.

### 5.2. Experimental Setup

The layout of the case study is based on a regional airport terminal, as seen in Figure 4. Five distinct regions are indicated: entrance area, check-in area, security checkpoint area, central shopping area, and gate area. The entrance area consists of 4 shops, 3 restaurants, and a restroom. The central shopping area consists of 5 shops, 3 restaurants, and 2 restrooms. The central shopping area does not contain areas where passengers can sit and wait. Thus, if passengers do not pursue activities in the central shopping area, they will move to the gate area.

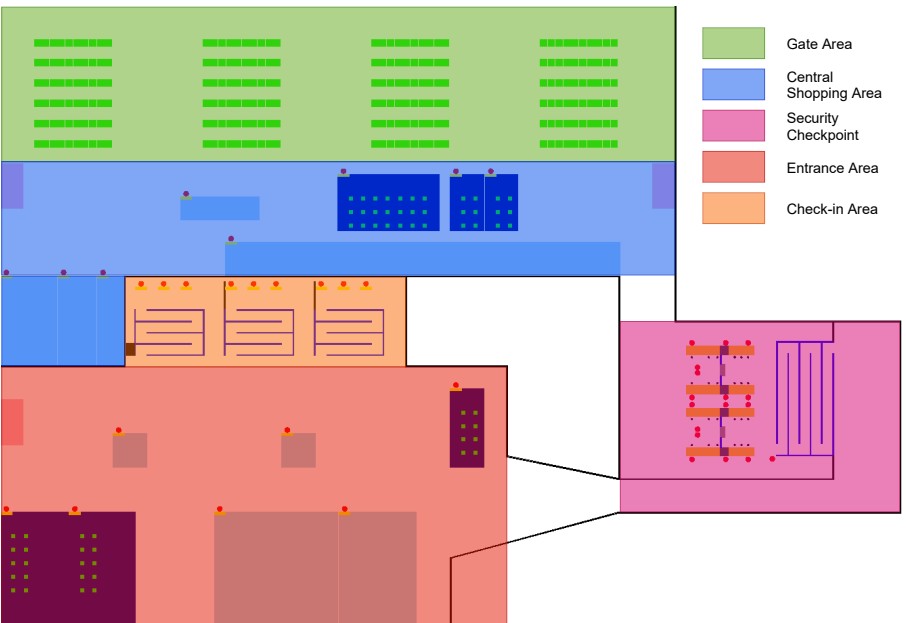

**Figure 4.** The environment of the agent-based model.

Table A1 in Appendix A contains relevant parameters for the simulated agent-based model. For each strategy or configuration of strategies, 1000 runs were performed. In the case study, there are three departing flights, departing at 16:30, 16:40, and 16:50. For each flight, 60 passengers depart. The simulation ran from 14:00 to 17:00. A missed flight in the simulation occurs when the passenger is not at the gate at flight departure.

### 5.3. Experiment 1: Effects of Time at Which Passengers Are Called to the Gate

For experiment 1, 10 call-to-gate strategies were used in which the time ranged from 30 min to 120 min with steps of 10 min. The call-to-gate below 30 min was not considered because the boarding process starts 30 min before flight departure. Strategy 2 from Table 1 was used as a fixed security lane allocation strategy in this simulation setting.

Results

For all call-to-gate strategies zero passengers miss a flight. This can be explained by multiple reasons. First, passengers arrive at the entrance at latest 30 min before flight departure, which for this scenario, it is always enough to reach the gate. Second, passengers react instantly if they are called to the gate, which in the experimental layout always leaves enough time to move from the central shopping area to the gate.

Figure 5 shows the total generated expenditure for different call-to-gate strategies. A call-to-gate time of 120 min leaves no dwelling time for passengers to make purchases, since passengers arrive at earliest 2.5 h before their flight. Between 100 and 80 min, the slope of the curve increases greatly. This can be attributed to airport enthusiasts who perceive that there is enough dwelling time to pursue activities that require time, such as restaurant and shop browsing activities. The slope transitions from increasing to decreasing at a call-to-gate time of 70 min. Since airport enthusiasts generate most expenditure, and 60% of airport enthusiasts arrive between 120 and 90 min before departure, it is important to allow this group to have enough dwelling time to pursue discretionary goals. Between 60 and 30 min call-to-gate time, most expenditure is generated by late arriving airport enthusiast passengers. The slope decreases because the majority of passengers have already completed their discretionary goals.

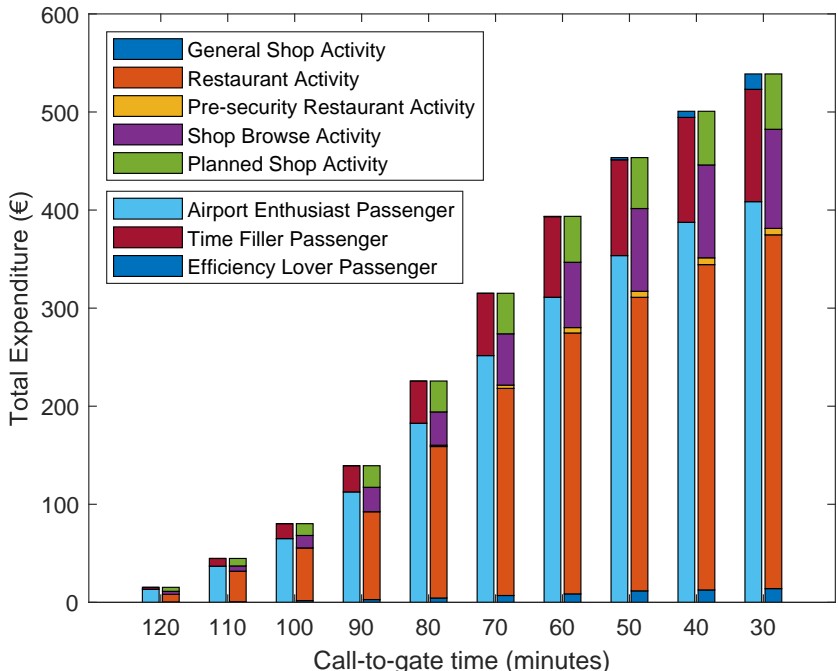

**Figure 5.** The impact of call-to-gate strategies on total generated expenditure. The indicated expenditure represents the mean value of 1000 simulation runs for that call-to-gate time. For each pair of bars, the left bar indicates the generated expenditure for each passenger type, the right bar indicates the generated expenditure for each activity.

Furthermore, the cognitive process of passenger agents was analyzed to gain more insight in the expenditure differences seen in Figure 5. Figure 6 shows the impact of the call-to-gate strategies on goal generation and completion. A goal is counted as completed if the passenger agent has interacted with the discretionary facilities. The number of goals for restaurant activities, pre-security restaurant activities, and planned shop activities, remains constant with different call-to-gate times. This is because the goals for these activities are generated probabilistically when the passenger agent is generated, which is explained in Section 4.3. For the shop browse activity this is also the case, but additionally the second decision-point results in an increase of generated shop browse goals when call-to-gate time decreases. The goals generated due to this second decision-point are always completed. More general shop goals are generated and completed when call-to-gate time decreases. The increase of completed restaurant goals contributes largely to the increased expenditure for lower call-to-gate times. The increase of generated and completed shop browse goals is also a large contribution to the expenditure. The shop browse generated expenditure is lower than the restaurant generated expenditure because of the assumed shop enter probability, and shop purchase probability of 0.2. Passenger agents engaging

with a restaurant will always make a purchase. Overall, most generated restaurant goals, shop browse goals, and planned shop goals transition from unachieved to completed when call-to-gate time decreases. This is because passenger agents have more free time, $T_f$, when call-to-gate time decreases. This causes passenger agents to perceive they have enough time to interact with discretionary activities, without increasing their perceived risk of missing a flight.

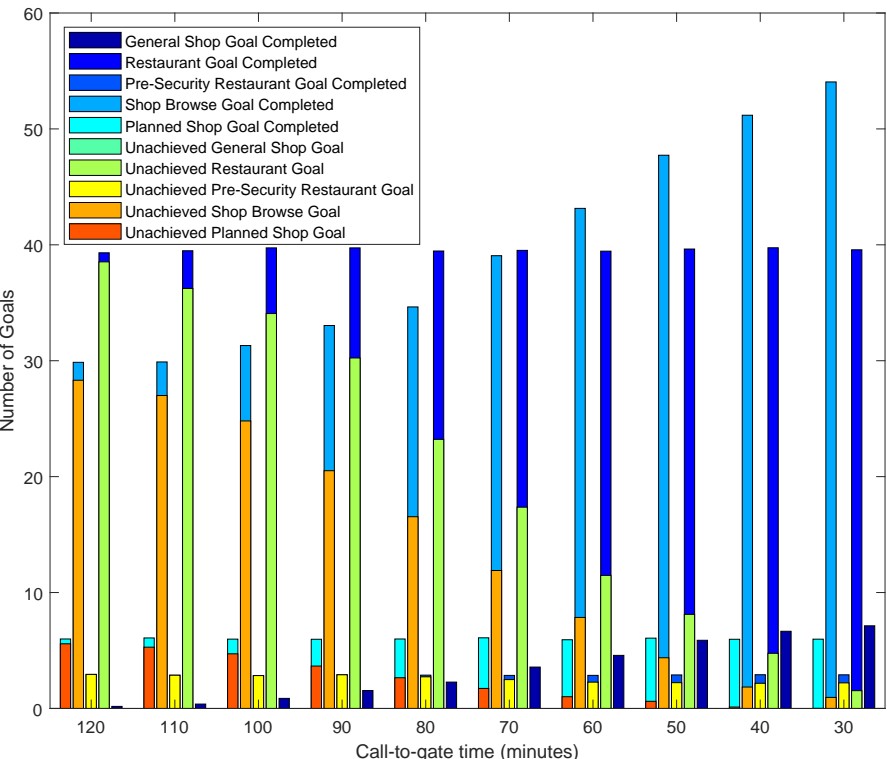

**Figure 6.** The impact of call-to-gate strategies on average number of generated passenger agent goals. The indicated number of goals represents the mean value of 1000 simulation runs for that call-to-gate time. The legend shows the distinction between completed and unachieved goals. Completed goals indicate that the passenger agents has interacted with the discretionary facilities.

In addition, the arrival time of passengers at the gate was analyzed. Figure 7 shows the difference between passenger gate arrival times of call-to-gate strategy of 30 and 120 min. For the call-to-gate strategy of 30 min, approximately 18 passengers out of 152 arrive later at the gate than the passengers in the call-to-gate strategy of 120 min. This can be attributed to the modeling choice at which passengers take a seat at their gate if they do not have any other goals. Since the majority of passengers do not interact with discretionary activities, call-to-gate strategies therefore do not affect their gate arrival. The gate arrival times of airport enthusiasts are impacted by call-to-gate times. More airport enthusiasts will arrive at a later time if the call-to-gate strategy is closer to 30 min.

Overall, call-to-gate strategy has a considerable impact on expenditure. For example, reducing call-to-gate time from 60 to 40 increases total expenditure by 27%. On average, each minute of extra dwelling time due to call-to-gate translates to an increase of EUR 5.8. Since there are no missed flights that occur for low call-to-gate times, these strategies are preferred to maximize expenditure and passenger goal completion.

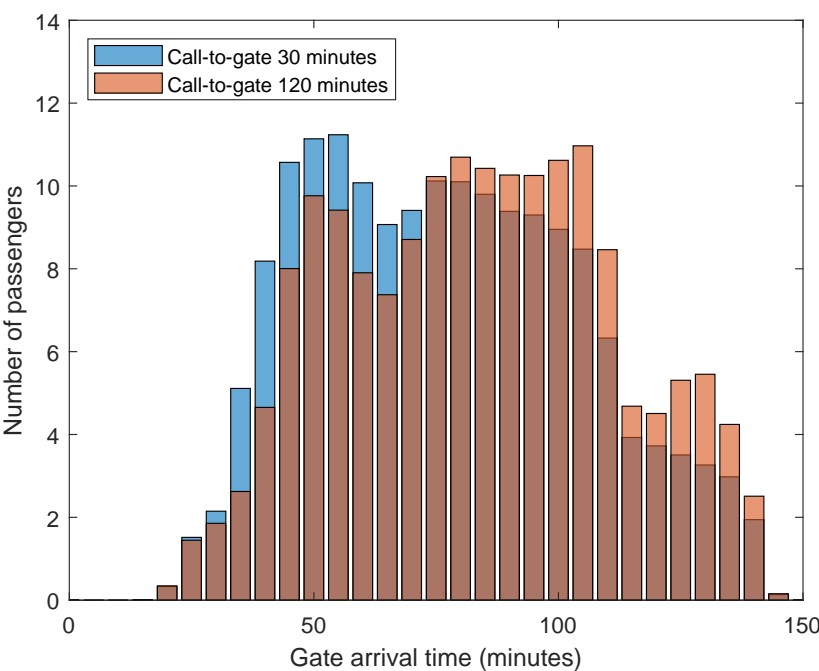

**Figure 7.** The average number of passenger agents that arrive at the gate, indicated in minutes before departure. The indicated number of passengers represents the mean value of 1000 simulation runs for that call-to-gate time.

### 5.4. Experiment 2: Effects of Security Lane Allocation

Table 1 shows the security lane allocation strategies that are used for experiment 2. It indicates how many security lanes are open per time interval. The choices for the strategies are based on the average number of passengers expected to arrive in each time interval. Between 14:30–15:30, more passengers are expected to arrive compared to what a single security lane is able to handle without accumulating queue time. A fixed call-to-gate strategy of 45 min is used for this experiment.

**Table 1.** Security lane allocation strategies, indicating how many security lanes are open per time interval.

| Strategy | 14:00–14:30 | 14:30–15:00 | 15:00–15:30 | 15:30–16:00 | 16:00–16:30 | 16:30–17:00 |
|:---:|:---:|:---:|:---:|:---:|:---:|:---:|
| 1 | 1 | 1 | 1 | 1 | 1 | 1 |
| 2 | 1 | 2 | 1 | 1 | 1 | 1 |
| 3 | 1 | 1 | 2 | 1 | 1 | 1 |
| 4 | 1 | 2 | 2 | 1 | 1 | 1 |

Results

Table 2 shows the relevant performance indicators for the lane allocation strategies and their values obtained by simulation. Strategy 4 is the best in terms of average queue time. Approximately 50 s of the security checkpoint queue time is spent moving through the queue, thus the actual waiting time of passengers in the queue for strategy 4 is very low. This implies that, for this strategy in the experiment, the total expenditure generated can not be further improved by alternative lane allocation strategies. Strategy 1 is the worst in terms of average queue time, approximately 5.6 min. Compared to strategy 4, the difference in generated expenditure is EUR 40.9, which is an average reduction of EUR 8.7 per minute of queue time. The reason for this is that a high number of airport enthusiasts become stuck in the queue, and may spend more than 10 min waiting. Strategy 2 and 3 perform much better than strategy 1 due to availability of an extra open lane for 30 min. Strategy 3 performs better than strategy 2, since the extra open lane between 15:00–15:30 is effective due to many passengers arriving in this interval. Furthermore, comparing strategies 1 and

2 with strategy 3, the difference between the expenditures is lower than the difference in security costs. It is interesting to note that 51.4 s difference in average queue time between strategies 2 and 3 corresponds to an expenditure difference of EUR 12.8, but between strategy 3 and 4, 42.9 s difference corresponds to a difference of only EUR 3.3. This implies that if we compare strategies 2 and 3 with strategy 4, the extra queue time for strategy 3 has a weaker impact on the planning and goals of passenger agents compared to strategy 2. Thus, average queue time is in itself not reliable to predict accurate changes in expenditure.

**Table 2.** Experiment 2 performance indicators.

| Strategy | Average Security Checkpoint Queue Time (s) | Total Expenditure (EUR) | Security Checkpoint Costs (EUR) |
|---|---|---|---|
| 1 | 337.6 | 428.7 | 360 |
| 2 | 149.1 | 453.5 | 420 |
| 3 | 97.7 | 466.3 | 420 |
| 4 | 54.8 | 469.6 | 480 |

### 5.5. Experiment 3: Combined Effect of Call-to-Gate and Security Lane Allocation

Experiment 3 uses the call-to-gate strategies from experiment 1, and the security lane allocation strategies from experiment 2. Thus, for each explored call-to-gate time, four security checkpoint lane allocation strategies are explored.

Results

Figure 8 shows the total generated expenditure dependent on call-to-gate strategies and the security lane allocation strategies. Strategies 1–4 correspond to the security lane allocation strategies from Table 1. For call-to-gate time from 120 to 100 min, generated expenditure does not depend on the strategies. Passengers are called to the gate too early for the average security queuing time to make a difference in expenditure. For call-to-gate time from 100 to 80 min, average security queue time increasingly affects total generated expenditure. In particular, at call-to-gate time of 80 min, strategy 1–4 generate EUR 203, EUR 227, EUR 217, and EUR 231, respectively. Thus, for this call-to-gate time, strategies 2 and 4 perform better because they have two security lanes in the 14:30–15:00 time interval. It is beneficial to allocate an additional lane in the 14:30–15:00 interval to process airport enthusiasts quicker such that they can interact with discretionary activities between 15:00–15:30. Allocating an extra lane in the 15:00–15:30 interval is less effective since these passengers will likely not have enough time to pursue a discretionary activity. This situation changes moving for call-to-gate time from 80 to 60 min. The gradient of the total expenditure for strategy 3 increases considerably compared to the other strategies, and from a call-to-gate time of 60 min performs better than strategy 2. The reason for this is that more passengers arrive at the security checkpoint between 15:00–15:30, compared to the 14:30–15:00 interval. This was concluded in experiment 2, which is why the average security queue time for strategy 3 was lower. However, with a call-to-gate time of 60 min, more airport enthusiasts will have the opportunity to engage with discretionary activities, compared to strategy 2. Moving from call-to-gate 60 to 30 min, the difference between strategies 3 and 2 further increases. Strategy 3 performs very close to strategy 4 for call-to-gate times of 40 and 30 min. This is because, for strategy 3, airport enthusiasts have enough time to complete all their discretionary goals.

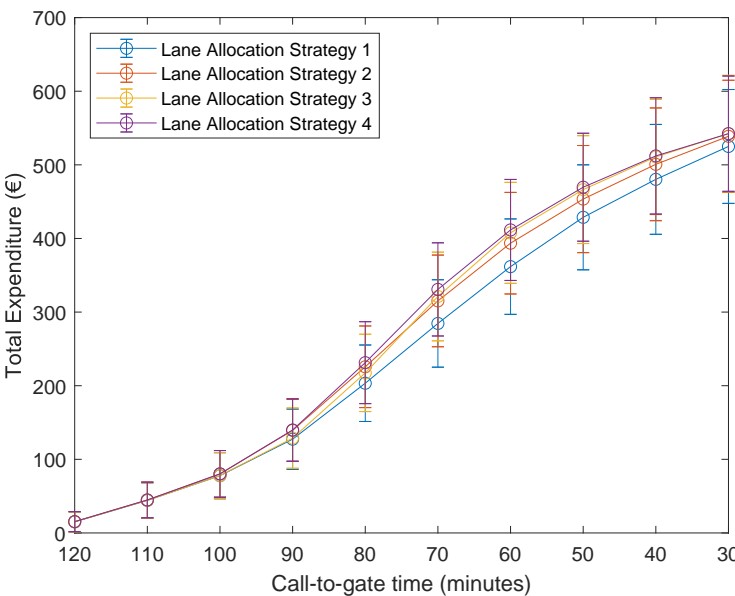

**Figure 8.** The impact of the 10 call-to-gate and 4 security lane allocation strategies on total generated expenditure. A dot represents the mean value of 1000 simulation runs for that strategy. The error bars indicate the standard deviation.

The results indicate that strategies 2 and 3, which are more cost-efficient due to lower resources, can perform as efficiently as strategy 4 in terms of total generated expenditure. For this experiment, a different combination of strategies may increase expenditure up to 16%. For example, at call-to-gate time of 70 min, strategy 1 generated EUR 285 and strategy 4 generated EUR 331.

## 6. Discussion

The agent-based simulation model in this study includes main airport terminal processes, discretionary activities, and a cognitive model for passenger decision-making with discretionary activities. The cognitive model is based on theories from behavioral sciences and allows for a more realistic approach to human decision-making at the airport terminal, compared to existing studies which include passenger interaction with discretionary activities [4,5,12]. The cognitive model for passenger decision-making with discretionary activities increases the complexity of the agent-based model, and additional data on passenger types, discretionary goal generation, emotions at the airport terminal, and decision-making with discretionary activities is required to calibrate the parameters of the cognitive model. More research will be required to identify aspects that influence passenger decision-making at the airport, and how the various emotions that passengers experience affect the decision-making.

The results from the experiments are airport specific, since for each airport, passengers engage differently with discretionary activities. The case study considers only three departing flights of 60 passengers and therefore provides a limited view of how operational strategies at the airport terminal can affect performance. The airport in the case study was based on an existing regional airport. Data from the regional airport were used to specify parameters related to arrival, check-in, and security. Assumptions were made on discretionary goal generation and decision-making.

In reality, operational call-to-gate strategies may result in boarding-incurred delays and missed flights. No missed flights were observed in the case study. This is because passenger agents have a constant accurate awareness of time, which allows them to react instantly when they are called to the gate. In reality, this could be different, and it depends on how passengers are called to the gate. For example, if passengers can follow on their mobile phones when the call-to-gate time is announced, they can react instantly just as in our model implementation. However, if the call-to-gate times are suddenly announced,

passengers could have a slower reaction time. Furthermore, the case study contains an airport terminal for which the central shopping area and gates are in close proximity. In practice, call-to-gate strategies are often employed in terminals with a pier configuration, in which the piers connect to the central shopping area. This forces passengers to stay in the central shopping area until the gate is known.

## 7. Conclusions and Future Work

The purpose of this study was to analyze how operational strategies for call-to-gate and security lane allocation influence passenger engagement with discretionary activities, and how that impacts airport terminal performance. An existing agent-based model of airport terminal operations, which covered main handling processes, was expanded to include decision-making about discretionary activities. To model passenger decision-making at the airport terminal, a cognitive model was developed, which was based on decision-making theories from behavioral sciences. The agent-based model was used to explore how operational strategies for call-to-gate influence expenditure and missed flights. Furthermore, effects of operational strategies for security lane allocation were investigated by analyzing average security checkpoint queue time, expenditure, and security checkpoint costs.

The results show that expenditure can significantly increase if passengers are called to the gate later. Calling passengers later to the gate increases dwelling time of passengers, which causes passengers to complete more discretionary goals. Airport enthusiast passengers who like to interact with discretionary activities are particularly important for generating expenditure.

Reducing average queue time at security checkpoints can also significantly increase expenditure, but the magnitude depends on the strategy. If the strategy reduces the dwelling time of airport enthusiast passengers, less goals involving discretionary activities are completed. If this is not the case, keeping more lanes open does not necessarily improve expenditure generation. Furthermore, call-to-gate strategies influence the effectiveness of security lane allocation strategies with respect to total generated expenditure. Call-to-gate strategies that result in little free time for passengers can benefit from additional security resources at time intervals at which early passengers arrive, since it can aid early passengers to complete some discretionary goals. However, call-to-gate strategies that result in more free time for passengers can benefit from additional security resources at time intervals when the majority of passengers arrive, since more passengers can complete discretionary goals.

Future work on this topic involves improving calibration and improving validation of discretionary activities within the model, and evaluating a case study with more data for passenger interaction with discretionary activities at the airport terminal. Additionally, passenger decision-making in the agent-based model can be improved by including more emotions in the cognitive architecture.

The model could also be used to further investigate interdependent relations between check-in desk allocation, security lane allocation, virtual queuing, and call-to-gate to improve airport terminal operational strategies. Since operational processes at the airport terminal are controlled by different stakeholders, the study could be extended to investigate how a multiple-criteria decision-making model can be built to improve airport terminal operational strategies by aligning the views of the stakeholders, maximizing their utility, or minimizing their disutility.

**Author Contributions:** Contribution of each co-author is distinguished as: Conceptualization, A.M., A.S., S.S.M.Z.; methodology, A.M., A.S. and S.S.M.Z.; software, A.M.; formal analysis, A.M.; writing—original draft preparation, A.M., A.S. and S.S.M.Z.; writing—review and editing, A.M., A.S. and S.S.M.Z.; visualization, A.M. and S.S.M.Z. ; supervision, A.S. All authors have read and agreed to the published version of the manuscript.

**Funding:** This research was funded by the 'European Regional Development Fund (ERDF) via the Kansen voor West II program' under the project 'Airport Technology Lab', grant number 'KVW-00235'.

**Institutional Review Board Statement:** Not Applicable.

**Informed Consent Statement:** Not Applicable.

**Data Availability Statement:** Not Applicable.

**Conflicts of Interest:** The authors declare no conflict of interest.

## Appendix A. Calibration

**Table A1.** Calibrated model parameters.

| Parameter | Value | Source |
|---|:---:|:---:|
| *Simulation parameters* | | |
| Number of runs per configuration | 1000 | - |
| Simulation start time | 14:00 | - |
| Simulation end time | 17:00 | - |
| Number of flights | 3 | - |
| Departure time (flight 1/flight 2/flight 3) | 16:30/16:40/16:50 | - |
| Passengers per flight | 60 | - |
| | | |
| *General model parameters* | | |
| Proportion passenger arrival | 0.95 | Assumption |
| Arrival distribution (early/middle/late) | 0.2/0.6/0.2 | Airport Data [42] |
| Arrival interval (AE, TF) | 9000–3600 s | Assumption [21] |
| Arrival interval (EL) | 7200–1800 s | Assumption [21] |
| Proportion passengers checked-in | 0.5 | Airport Data [42] |
| Check-in time | $N(60, 6)$ s | Airport Data [42] |
| Luggage drop time | $N(65.9, 36.3)$ s | Airport Data [43] |
| Luggage collect time | $N(60.5, 52.4)$ s | Airport Data [43] |
| Luggage check time | $N(59.3, 52.5)$ s | Airport Data [43] |
| Proportion passenger physical check | 0.0916 | Airport Data [43] |
| Physical check time | $N(22.5, 18.2)$ s | Airport Data [43] |
| Proportion X-ray threat | 0.0750 | Airport Data [43] |
| | | |
| *Trait model parameters* | | |
| Passenger type distribution (AE/TF/EL) | 0.35/0.48/0.17 | Airport Data [21] |
| Age distribution ($\leq$20/21–30/31–40/41–50/51–60/$\geq$60) | 0.013/0.194/0.283/0.251/0.164/0.095 | Airport Data [41] |
| Proportion gender (m/f) | 0.5/0.5 | Assumption |
| Proportion class (leisure/business) | 0.9/0.1 | Assumption |
| Proportion passenger waver | 0.12 | Airport Data [19] |
| | | |
| *Discretionary goal generation parameters* | | |
| Restaurant proportion (AE/TF/EL) | 0.5/0.1/0.1 | Assumption |
| Pre-security restaurant proportion (AE/TF/EL) | 0.5/0/0 | Assumption |
| Planned shop proportion (AE/TF/EL) | 0.1/0/0 | Assumption |
| Shop browse proportion (AE/TF/EL) | 0.5/0/0 | Assumption |
| Shop browse proportion (2nd chance) (AE/TF/EL) | 0.4/0.2/0.1 | Assumption |
| General shop proportion (AE/TF/EL) | 0.01/0.01/0.01 | Assumption |
| Restroom probability (per min) | $\frac{1}{180}$ | Assumption |

**Table A1.** *Cont.*

| Parameter | Value | Source |
|---|---|---|
| *Decision-making model parameters* | | |
| Restaurant $T_f$ threshold | $T_f > 10$ min | Assumption |
| Restaurant duration | $\Gamma(5.62, 3.93)$ min | Assumption [18] |
| Pre-security Restaurant $T_f$ threshold | $T_f > 30$ min | Assumption |
| Pre-security restaurant duration | $\Gamma(5.62, 3.93)$ min | Assumption [18] |
| Planned shop $T_f$ threshold | $T_f > 10$ min | Assumption |
| Planned shop duration | $N(4, 1)$ min | Assumption |
| Shop browse $T_f$ threshold | $T_f > 10$ | Assumption |
| Shop browse (2nd chance) $T_f$ threshold | $T_f > 30$ | Assumption |
| Shop browse duration | $\Gamma(5.76, 4.76)$ min | Assumption [18] |
| General shop $T_f$ threshold | $T_f > 10$ min | Assumption |
| General shop duration | $N(4, 1)$ min | Assumption |
| Restroom duration (m/f) | $LogN(4.70, 0.61) / LogN(4.94, 0.59)$ min | Airport Data [44] |
| Shop browse shop enter probability | 0.2 | Assumption |
| Shop purchase probability | 0.2 | Assumption |
| Shop purchase expense | $\Gamma(5.0, 2.0)$ € | Assumption |

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
