# Peer review of "Systemic Agent-Based Modeling and Analysis of Passenger Discretionary Activities in Airport Terminals"

_aerospace, doi:10.3390/aerospace8060162_

Round 1
Reviewer 1 Report
Three considerations:
- Validation is a must, not a future work.
- Plus. Security cost is not linear. Adding new lines or increasing rate efficiency is costly. Usually, to extend the number of security lines at the airport requires big works in operating terminals.
- Minor. You work with anxiety and you talk about emotions. I suggest to talk exclusively about anxiety (related with call-to-gate time and the possibility to miss a flight). There are other emotions that are important when we analyze passengers behaviour. for example, the passengers' satisfaction with the experience at different airport processes (i.e. security process is a violent experience for many people). May be it could be introduced an indicator to consider this kind of qualitative experiences.
Author Response
Dear reviewer,
We would like to thank you again for having read our paper and providing your valuable comments. We tried to address your comments as much as possible in terms of contents when the time limit of 10 days allowed us. See below for detailed answers of the comments of the reviewers.
- Validation is a must, not a future work.
Parts of the model (check-in and security process) were calibrated on real-life passenger data from the considered regional airport, such that the model represents reality. Additionally, AATOM features such as check-in and security processes have been validated in AATOM case studies conducted by previous researchers. We have explicitly added this information in the calibration section, 4.10.
In our research, full model validation would require all model parameters to be calibrated to data from a single airport. This requires a lot of data/knowledge of the airport terminal, e.g., how different types of passengers spend time and money in each specific shop/restaurant, what passengers goals are when they arrive at the airport, and how the airport environment and their perceptions may influence passenger decisions. We have had interviews with commercial managers at the airport formulate our assumptions regarding passengers interaction with discretionary activities. Accurate data on this was not available to us, and therefore full validation of passenger interaction w.r.t discretionary activities, was very difficult. We have added a part on this at the bottom of the calibration section. Furthermore, we have rewritten the future work part in the conclusion to state that validation needs to be improved w.r.t to discretionary activities.
2. security cost is not linear. Adding new lines or increasing rate efficiency is costly. Usually, to extend the number of security lines at the airport requires big works in operating terminals.
We have added extra clarification that the security checkpoint costs, for experiment 2, are represented by the security operator costs.
Because we are dealing with security lane allocation (of already available lanes), security lane operating costs mainly determine the security cost.
We’ve had interviews with a security expert from a regional airport, and besides operating cost for different number of lanes, no other security costs were mentioned to be of importance.
3. Minor. You work with anxiety and you talk about emotions. I suggest to talk exclusively about anxiety (related with call-to-gate time and the possibility to miss a flight). There are other emotions that are important when we analyze passenger behaviour. For example, the passengers’ satisfaction with the experience at different airport processes (i.e. security process is a violent experience for many people). May be it could be introduced an indicator to consider this kind of qualitative experiences.
The airport terminal is an environment in which passengers experience many different emotions, with changing intensities. The passengers’ cognitive architecture in the agent-based model allows for more emotions, besides anxiety, to be implemented. However, it is very difficult to anticipate on how all possible emotions would impact decision-making with respect to discretionary activities. For example, how would positively valenced emotions such as joy/excitement/satisfaction impact passenger decision-making?
Research needs to be conducted to better understand this.
For anxiety, the impact on the decision-making is more clear and straight-forward. Passengers can remove discretionary activities from their planning because they are not mandatory, and they will do so if they think the activity introduces a risk to catching their flight.
The 2nd paragraph of the ‘passenger agent emotions’,. section, 4.8, was altered to make our point more clear. Additionally, including more emotions is now stated in the future work, in section 7.
Reviewer 2 Report
The paper is well structured and addresses, I suppose, a relevant issue for terminal managers. Being a frequent flyer and user of terminals, economist, and acquainted with agent based modeling, but certainly not an expert in this particular field, I find the paper easy to understand. The only small point I would like to make is that the results seem to be so intuitive that it seems hardly necessary to develop an agent based model (that perse does not contribute anything to our knowledge of agent based modeling as far as I can oversee) to arrive at them.
Author Response
Dear reviewer,
We would like to thank you again for having read our paper and providing your valuable comments.
Although there are some parts that, of course, are intuitive, we have also analyzed how operational strategies for call-to-gate and security lane allocation influence passenger engagement with discretionary activities, and how that impacts airport terminal performance. The agent-based model was used to explore how operational strategies for call-to-gate influence expenditure, and missed flights. Furthermore, effects of operational strategies for security lane allocation were investigated by analyzing average security checkpoint queue time, expenditure, and security checkpoint costs. Moreover, a cognitive model was developed which was based on decision-making theories from behavioral sciences.
Reviewer 3 Report
A very interesting research. The methodology used is clearly presented and the Agent-based modelling was correctly used. The conclusion are supported by the results obtianed.
Author Response
Dear reviewer,
Thanks a lot for reading our paper and we are very much happy that you are satisfied with our paper.
Round 2
Reviewer 1 Report
ok. Thank you for your comments.